# Construction and Practice of Livelihood Efficiency Index System for Herders in Typical Steppe Area of Inner Mongolia Based on Super-Efficiency Slacks-Based Measure Model

**Gerile Qimuge [1], Wulan Tuya [1,2,3,*], Si Qinchaoketu [1,2,3] and Bu He [1]**

1   College of Geographical Science, Inner Mongolia Normal University, Hohhot 010022, China; grlqmg1020@163.com (G.Q.); chogoo2020@imnu.edu.cn (S.Q.); buhe15540767179@163.com (B.H.)
2   Key Laboratory of Disaster and Ecological Security on the Mongolian Plateau, Inner Mongolia Autonomous Region, Hohhot 010022, China
3   Key Laboratory of Climate Change and Regional Response on the Mongolian Plateau, Autonomous Region, Hohhot 010022, China
*   Correspondence: mtuya1967@163.com

**Abstract:** Inner Mongolia is one of the main animal husbandry production bases in China, with herders being the main animal husbandry producers. A systematic analysis of the efficiency of herding households' livelihoods and the influencing factors is of great importance to formulate effective policies to support herding households' livelihoods, enhance their social adaptability, and alleviate the vulnerability of poor people in herding areas. This study used a typical steppe of Inner Mongolia as the research area. It used the interview data of herding households from 2021, constructed the evaluation index system of herding households' livelihood efficiency, analyzed the redundancy of the inputs and outputs of herding households' livelihoods, and examined the key factors affecting herding households' livelihood efficiency. The results indicate that (1) the pure technical effectiveness of the livelihood efficiency of typical grassland herding households in Inner Mongolia is the highest; the comprehensive technical efficiency and scale efficiency are low. The scale return of most herders' livelihoods shows a decreasing state. (2) According to the results of the model, under the premise of the output not being reduced, reducing the amount of social capital input can effectively save resources. Without increasing the input, the room for improvement in the living level is the most obvious. (3) The pasture area, the communication network, and the access to information have significant negative effects on the efficiency of herders' livelihoods; infrastructure and water supply have significant positive impacts. In summary, we built a model for evaluating the livelihood efficiency of herders in typical grassland areas of Inner Mongolia, which can provide a reference for the revitalization work of pastoral areas and related research in the future.

**Keywords:** farmer household livelihoods; livelihood efficiency; revitalization of pastoral areas; Mongolian Plateau





## 1. Introduction

Traditional herders' lives are based on extensive grassland grazing. After the 1980s, the policy of "two rights and one system" and the compensation policy for grassland ecological protection were gradually implemented. The herders in Inner Mongolia switched from nomadic herding to settled herding, and the grazing mode changed from grassland grazing all year round to grassland grazing combined with enclosure grazing. Under multiple pressures, such as the change in the grazing mode, a continuous warm and dry climate (He et al., 2022) [1], and the increase in population, the contradictions between people and land, and between grass and livestock in pastoral areas, have become increasingly prominent. The three herding problems (Li et al., 2013) [2], mainly characterized by grassland degradation, rising animal husbandry costs, and the livelihood difficulties of

herders, have become increasingly serious. Thus, the sustainable development of pastoral areas faces severe challenges. Balancing the contradiction between people and land, solving the livelihood difficulties of herders and realizing the sustainable development of pastoral areas are hot topics in the field of grassland research.

Current research on the livelihood of herders focuses mainly on its sustainability (He et al., 2022) [3], vulnerability (Li et al., 2023) [4], and resilience (Hou et al., 2012) [5]; grassland subsidy policy results (Ding et al., 2022) [6]; the transfer of herders' grassland (Zhang et al., 2017) [7]; and livelihood strategy selection (Su et al., 2022) [8]. There are few studies on the livelihood capital inputs and outputs of herding households, that is, their livelihood efficiency, and the research mainly focuses on the effects of grassland transfer and grassland protection compensation policies on herding households' production efficiency (Shi et al., 2022; Wang et al., 2021) [9,10]. The concept of livelihood efficiency was first proposed by Su Fang in 2021 [11]. The efficiency of a herding household's livelihood is the ratio of the resources invested in the herding household's livelihood activities to the output obtained. This can indicate the allocation state, the utilization effect, and the management decision-making level of the input capital elements in the livelihood activities of herders. Scholars have conducted a lot of research on the livelihood and production efficiency of farmers and herders. For example, Twumasi et al. (2021) [12] and Mezgebo et al. (2021) studied the factors that influence farmers' resource use efficiency in Ethiopia [13]. Cobbinah et al. (2023) evaluated the impact of farmers' mutual labor support on their productivity and technical efficiency [14]. In fact, the livelihood efficiency of farmers and herders is affected by many factors. In the current study, the efficiency evaluation is limited to a specific aspect of livelihood activities, and there is a lack of a comprehensive evaluation of all livelihood activities. In most research, the livelihood capital is usually used to measure the results of farmers' and herders' livelihood activities, the ability to achieve sustainable development, and the ability to resist risks (Wang et al., 2019) [15], while ignoring the subjective initiative of farmers and herders in utilizing the livelihood capital. As an input factor for farmers and herders, the livelihood capital cannot comprehensively measure the effects of the livelihood activities of farmers and herders (Wang 2018) [16]. Therefore, in the calculation of the livelihood efficiency, the livelihood capital is used as the input factor for livelihood when trying to evaluate the overall livelihood efficiency of herders.

A data envelopment analysis (DEA) evaluates the relative effectiveness based on multi-input/multi-output data, and was proposed by Charnes in 1978 [17]. As the efficiency value has the feature of data truncation, scholars use the Tobit model to avoid the problem of efficiency value limitation (Tobin 1958) [18]. This method, combined with DEA and the Tobit regression model, has been widely used by scholars in studies on agricultural production (Gul et al., 2009) [19], energy (Ervural et al., 2018) [20], the ecological environment (Wang et al., 2021) [21], green development (Yang et al., 2023) [22], the medical system (Cheng et al., 2022) [23], enterprise operations (Wei et al., 2023) [24], and employee work efficiency (Otero et al., 2012) [25]. The super-efficiency slacks-based measure (SE-SBM) model is a common model used in DEA methods, which can also calculate the part with an efficiency value greater than one. Scholars have conducted a lot of research using this model. For example, Shah et al. (2022) explored the impact of non-performing loans on the operational efficiency of commercial banks in Pakistan [26]. Khan et al. (2022) investigated the impact factors of the rural sustainable development efficiency in the Yellow River Basin from 1997 to 2017 [27]. Huang et al. (2023) measured the spatial and temporal variations of the ecological efficiency of Zhejiang Province in China [28]. Nguyen et al. (2023) evaluated the foreign direct investment attractiveness of Vietnamese provinces from 2017 to 2021 [29].

It can be seen that there are few research studies on the livelihood efficiency of herders in the current pastoral area of Inner Mongolia. Under the double pressure of environmental and social changes, it is particularly important to correctly understand the inputs and outputs of herdsmen's livelihood production activities to improve their quality of life. This study takes typical steppe herders in Inner Mongolia as the research objects, uses the interview data of herders in 2021, and establishes an evaluation index system for the

livelihood efficiency of typical steppe herders in Inner Mongolia with reference to relevant materials and research. The SE-SBM model was used to evaluate the livelihood efficiency level of typical steppe herders in Inner Mongolia. On this basis, the improvement plan of herders' livelihood inputs and outputs and the factors affecting their livelihood efficiency were further analyzed. This provides a reference for improving the livelihood level of herders and revitalizing herding areas in Inner Mongolia.

## 2. Materials and Methods

### 2.1. Study Area

We used typical steppe areas in Inner Mongolia as the study areas, and these are located in East and West Ujumchin Banner in the Xilingol League and Xinbarhu Left Banner in Hulunbuir, Inner Mongolia Autonomous Region (Figure 1). The main landform types are high plains and hills, high in the southeast and low in the northwest, with an average elevation of 829 m. The study area has a temperate continental monsoon climate, with long and cold winters and short and mild summers. According to meteorological station data from 1960 to 2019, the average annual temperature in the study area was 1.2 °C, and the average annual precipitation was approximately 285 mm. The precipitation was mainly concentrated in July and August, and the rain and heat were present during the same period. The vegetation type is typical steppe, the soil type is mainly chestnut soil, and the hidden soil includes light chestnut, dark chestnut, and aeolian sand soils. Surface runoff is not developed, and the main rivers and lakes are the Baragar River, Gaolihan River, and Hulun Lake.

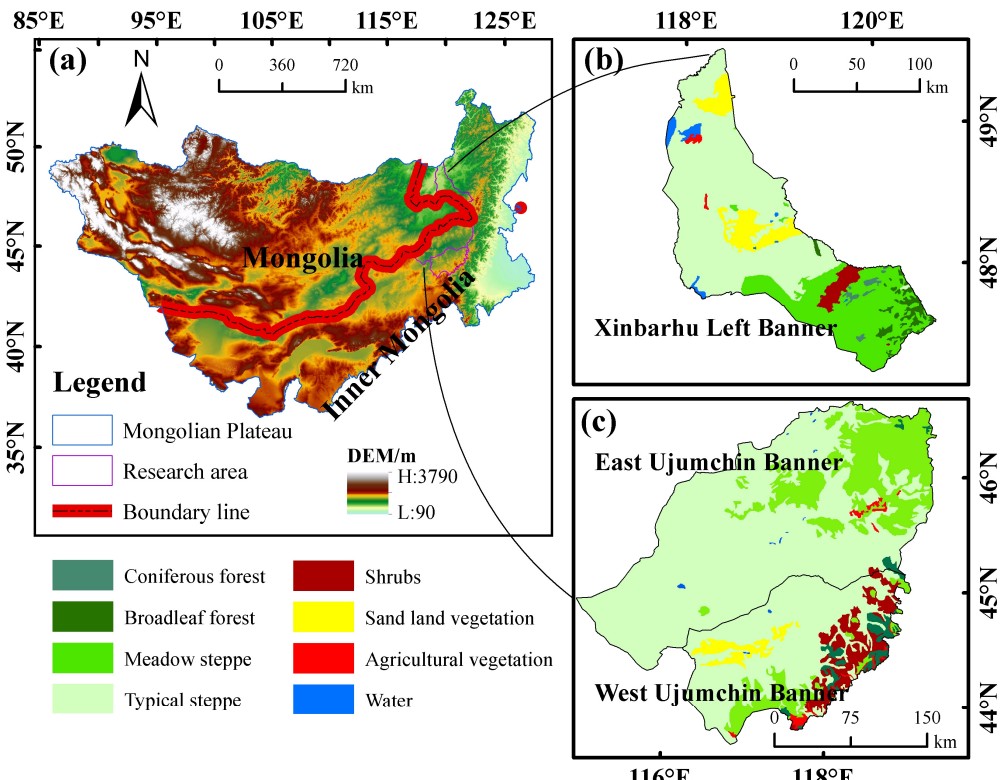

**Figure 1.** Location of study area. (**a**) Mongolian Plateau; (**b**) Xinbarhu Left Banner in Hulunbuir; (**c**) East and West Ujumchin Banner in the Xilingol League.

The total land area of East and West Ujumchin Banner is 70,100 km². According to statistics for 2022, the permanent population of the two Banners was 157,700, and the pastoral area accounts for 56.4% of the total population. The gross domestic product (GDP) of the two Banners was CNY 28.808 billion, of which the total output value of animal husbandry was CNY 4.734 billion, accounting for 16.43% of the regional GDP, and the total

output value of the secondary industry was CNY 18.523 billion, accounting for 64.30% of the regional GDP. The annual per capita disposable income of permanent residents in the two Banner towns and agricultural and pastoral areas was CNY 41,400.

The total land area of Xinbarhu Left Banner is 20,100 km$^2$. According to the statistics in 2022, the permanent population was 41,500, and the population in pastoral areas accounts for 77.74%. In 2021, the GDP of the region was CNY 2.701 billion, of which the total output value of animal husbandry was CNY 1.219 billion, accounting for 45.13% of the GDP, and the total output value of the tertiary industry was CNY 1.075 billion, accounting for 39.78% of the GDP. The annual per capita disposable income of permanent residents in agricultural and pastoral areas is CNY 27,200.

*2.2. Data Sources*

The data in this study were obtained from interviews in June and July 2021 with contracted herders of the grassland where the sampling points are located, with a total of 90 copies. There were 36 in the East and West Ujumchin Banner, and 54 in Xinbarhu Left Banner. A questionnaire survey and in-depth interviews were conducted. The main interviewees were household heads, and the interview lasted 1–3 h. The content of the investigation included (1) basic family information, such as name, age, education level, working ability, and health status; (2) herders' living conditions and social relationships, such as electricity, water, food security, communication network, relatives and friends, policy concerns, education and medical conditions, entertainment, knowledge and skills related to animal husbandry, ecological protection consciousness, and satisfaction with life; (3) assets of herders, such as pasture area, number of livestock, and housing and production equipment; and (4) the income of pastoral households, such as the types and amounts of income, state subsidies, and bank loans. Due to the low population density in pastoral areas, the number of herders interviewed was relatively small, but the herders' living patterns in the study area were relatively similar. Therefore, the investigated samples could reflect the livelihood of herders in the study area.

*2.3. Research Methods*

2.3.1. The Construction of Index System

Livelihood efficiency is the ratio between the livelihood resource input and livelihood output obtained using herders' livelihood activities, which is measured by two parts: livelihood capital and livelihood output. The livelihood capital included in the sustainable livelihood analysis framework established by the British Agency for International Development is divided into five categories—human, physical, natural, financial, and social capital [30]. Odero puts forward information capital as the sixth type required for farmers' livelihood. As the information network continues to develop at a high speed, information capital is also an indispensable part of herders' livelihoods. Therefore, the above six types of livelihood capital are taken as input variables for herders' livelihoods in this study. With reference to relevant studies [11,31,32], the income level, the living level, the welfare level, the entertainment richness, and the happiness of herders were selected as variables of livelihood output of herders, which can be represented to some extent. Twenty indices of input and output variables were selected to construct an evaluation index system of livelihood efficiency of typical steppe herders in Inner Mongolia (Table 1).

**Table 1.** Evaluation index of livelihood efficiency of herding households.

| Evaluation Indices | | Variable | Definition |
|---|---|---|---|
| Livelihood input | Human capital | Education | Education level: college or above = 1; senior high school = 0.75; junior high school = 0.5; primary school = 0.25; illiterate = 0 |
| | | Labor force | Number of household labor force |
| | | Labor capacity | The overall labor capacity of the household: 18~60 years old = 1; 12~18 years old and 60~70 years old = 0.5; over 70 years old, under 12 years old, and those who are unable to work = 0 |
| | Physical capital | Livestock | Number of livestock: calculated in sheep units, horse = 6; cow = 5; sheep = 1; goat = 0.8 |
| | | House | Dwelling house: brick house = 3; steel house = 2; adobe house or yurt = 1; no house = 0. House area: >120 m$^2$ = 1; 90 to 120 m$^2$ = 0.75; 60 to 90 m$^2$ = 0.5; 30 to 60 m$^2$ = 0.25; <30 m$^2$ = 0 |
| | | Machinery | Farm machinery and means of transportation owned by herdsmen: cars or large farm machines (>36 kW/h) = 3; medium-sized farm machines (18~36 kW/h) = 2; motorcycles or small farm machines (<18 kW/h) = 1; none = 0 |
| | Natural capital | Pasture area | Actual pasture area (hm$^2$): contracted pasture area + rent-in area—rent-out area |
| | | Pasture quality | Good = 3; normal = 2; bad = 1 |
| | Financial capital | Cash income | Per capita income (CNY 10,000): >10 = 5; 5~10 = 4; 2~5 = 3; 1~2 =2; <1 = 1 |
| | | Loan | Loan amount (CNY 10,000): >10 = 2; ≤10 = 1; no = 0 |
| | Social capital | Social network | Professional types of contacts: cadre = 4; merchant = 3; worker = 2; herder or farmer = 1 |
| | | Access to help in times of hardship | Number of channels to receive help |
| | Information capital | Whether there is a communication network | Yes = 1; no = 0 |
| | | Access to information | Number of channels to obtain information |
| | | Whether the access to information is timely | Yes = 1; no = 0 |
| Livelihood output | Income level | Total cash income | >50 = 5; 20~50 = 4; 10~20 = 3; 5~10 = 2; <5 = 1 (unit: CNY 10,000) |
| | Living level | Food and water security | Very high = 5; high = 4; normal = 3; low = 2; very low = 1 |
| | Welfare level | Education and health | Very good = 5; good = 4; normal = 3; bad = 2; very bad = 1 |
| | Entertainment richness | Number of recreational activities attended | Very much = 5; more = 4; normal = 3; less = 2; very few = 1 |
| | Happiness | Degree of love for life and perception of happiness | Very high = 5; high = 4; normal = 3; low = 2; very low = 1 |

### 2.3.2. Explanatory Variable Selection

The livelihood behavior of herders may be influenced by the characteristics of the family (Yan 2010) [33], society (Xiong et al., 2021) [34], and natural environment (Liu 2021) [35]. In addition to the variables already included in the index system, several indicators that may affect the livelihood of herders were added. First, in terms of family

characteristics, we selected two factors: animal husbandry knowledge and skills and ecological protection consciousness. The degree of knowledge and skill mastery required for animal husbandry directly affects the livelihood and production of herders. The higher the degree of knowledge and skill mastery, the fewer problems the herders encounter in production activities, and the higher the production efficiency. The strength of the ecological protection consciousness of herders is related to their production activities. To obtain more output, herders may, regardless the state of the pasture, raise more livestock if they have weaker ecological protection consciousness. Second, in terms of social characteristics, factors such as infrastructure and medical security are conducive to compensating for the shortcomings of livelihood capital and optimizing the livelihood mode of herders (Wang et al., 2019) [36]. Therefore, two variables, infrastructure and medical security, were selected as social factors in this study. Finally, in terms of the characteristics of the natural environment, the two main natural resources that herders rely on for their livelihood are water sources and grasslands. Water supply and grassland ecological conditions directly affect the sustainability of livestock production and the livelihoods of herders. Pasture quality has been included in the index system, so the water supply is considered a natural environmental factor. The above factors were added as explanatory variables to determine their effects on the livelihood efficiency of herders. These variables are presented in Table 2.

**Table 2.** Supplementary explanatory variables for the impact of herders' livelihood efficiency.

| Variable | Definition | Maximum Value | Minimum Value | Mean Value | Standard Deviation |
|---|---|---|---|---|---|
| Animal husbandry knowledge and skills | The mastery of animal husbandry knowledge and skills 1–5 | 5 | 1 | 4.29 | 0.82 |
| Ecological protection consciousness | The degree of environmental protection consciousness 1–5 | 5 | 2 | 4.19 | 1.03 |
| Infrastructure | The degree of infrastructure improvement 1–5 | 5 | 1 | 3.99 | 0.91 |
| Medical security | The degree of medical security 1–5 | 5 | 1 | 4.07 | 0.98 |
| Water supply | Accessibility of water 1–5 | 5 | 3 | 4.19 | 0.47 |

### 2.3.3. Sample Characteristics

Descriptive statistical analysis was performed on the basic characteristics of the herders (Table 3). The age of participants was mostly between 30 and 60 years, accounting for 56.36% of the total sample of herders. The education level of the surveyed herders was generally not high, with 41.04% having primary school or below. The dependency ratio of the surveyed herders was medium, and 48.89% of them had a dependency ratio between 0.5 and 1. The annual per capita income of most herders surveyed was more than CNY 10,000, and those with per capita incomes of CNY 10,000–50,000 were the largest, accounting for 35.56% of the total sample.

**Table 3.** Characteristics of interviewed herders.

| Index | Category | Percentage | Index | Category | Percentage |
|---|---|---|---|---|---|
| Age | <30 years old | 34.39% | Dependency ratio | <0.5 | 34.44% |
| | 30~60 years old | 56.36% | | 0.5~1 | 48.89% |
| | >60 years old | 9.25% | | >1 | 16.67% |
| Education level | Primary and below | 41.04% | Per capita income (CNY 10,000) | ≤1 | 7.78% |
| | Junior high school | 27.75% | | 1~5 (include 5) | 35.56% |
| | Senior high school | 11.27% | | 5~10 (include 10) | 25.56% |
| | College or above | 19.94% | | >10 | 31.11% |

2.3.4. Livelihood Efficiency Evaluation Model

(1)  SE-SBM model

Traditional DEA methods are radial and angular models. When there is over-input or under-output, that is, non-zero slack of input or output, radial DEA will overestimate the efficiency of the Decision-Making Unit (DMU). However, the Angle DEA must ignore the change in input or output, and the calculated results are not in line with the objective reality (Fare et al., 2010) [37]. The SE-SBM model is an efficiency measurement method based on relaxation variables. If the calculated efficiency value is >1, it is regarded as super efficiency, and if it is <1, it is regarded as an invalid state (Tone 2002) [38].

In this study, the SE-SBM model of data envelopment analysis was used to calculate the livelihood efficiency of herders. First, it is assumed that the return to scale is constant, that is, the livelihood efficiency of herders with fixed input and fixed output is measured using the comprehensive technical efficiency (TE). Second, it is assumed that the variable returns to scale, that is, the livelihood efficiency of herders with maximum output at fixed inputs is measured with pure technical efficiency (PTE). TE can be decomposed into the PTE product and scale efficiency (SE). The change in SE reflects the impact of input growth on productivity. According to SE, the livelihood efficiency of herders is in the range of increasing or decreasing returns to scale, so that the livelihood production scale of herders can be adjusted to reach the best production frontier.

First, the range standardization method is used to standardize the input index data to eliminate the difference of different dimensions and orders of magnitude. Second, the entropy method is used to determine the weight of each input index, excluding the influence of subjective factors. Then, the comprehensive weighted average method is used to calculate the livelihood input of herders. Finally, based on the input and output data, the following formula is used to calculate the livelihood efficiency of the sample herders:

$$\rho = min \frac{\frac{1}{m}\sum_{i=1}^{m}\frac{\overline{x}_i}{x_{i0}}}{\frac{1}{s}\sum_{k=1}^{s}\frac{\overline{y}_k}{y_{k0}}} \tag{1}$$

$$s.t. \ \overline{x}_i \geq \sum_{j=1,\neq 0}^{n}\lambda_j x_j, \forall i; \tag{2}$$

$$\overline{y}_k \leq \sum_{j=1,\neq 0}^{n}\lambda_j y_j, \forall k; \tag{3}$$

$$\overline{x}_i \geq x_{i0}, \ 0 \leq \overline{y}_k \leq y_{k0}, \lambda_j \geq 0, \sum_{j=1,\neq 0}^{n}\lambda_j = 1, \forall i, j, k; \tag{4}$$

where $\rho$ represents the super-efficiency value, $x$ represents the input index, $y$ represents the output index, $n$ represents the number of DMU, $\overline{x}$ and $\overline{y}$ represent the relaxation variables of the input and output, respectively, $m$ and $s$ represent the number of variables of the input and output, respectively, and $\lambda_j$ represents the weight of the $j$th DMU.

(2)  Tobit Regression Model

The Tobit regression model, proposed by James Tobin in 1981, is based on the maximum likelihood estimation method, which can better-avoid the problems of parameter inconsistency and bias. Therefore, a truncated Tobit regression model with limited dependent variables was adopted to analyze the influencing factors. The specific model form was set as

$$\rho = \begin{cases} \rho^* = \alpha_0 + \sum_{j=1}^{l}\alpha_j\beta_{ij} + \varepsilon_i, 0 \leq \rho^* \leq 1 \\ 0, \rho^* < 0 \\ 1, \rho^* > 1 \end{cases} \tag{5}$$

where $\rho$ represents the explained variable, $\rho^*$ represents the latent variable, $\alpha_0$ represents the constant term, $\alpha_j$ represents the regression coefficient of the explanatory variable, $\beta_{ij}$ represents the explanatory variable, and $\varepsilon_i$ represents the random error term.

## 3. Results

### 3.1. Livelihood Efficiency of Typical steppe Herders in Inner Mongolia Pastoral Area

The efficiency of herders' livelihoods reflects the output of their livelihood capital invested in their livelihood activities. If the efficiency value reaches 1, it indicates that the livelihood efficiency of the herder has reached optimal efficiency, and if it is <1, it indicates that the livelihood efficiency of the herder has not reached optimal efficiency. Because there are many herders with an efficiency value of one, the SE-SBM model is used to calculate the efficiency value of the herder to better-distinguish the livelihood efficiency of the herder. The livelihood efficiency of typical steppe herders in the Inner Mongolia pastoral area, including TE, PTE, and SE, is shown in Figure 2. With reference to the classification standards of efficiency values in relevant studies (Wei et al., 2011) [39], in this study, the livelihood efficiency of pastoral households is divided into five levels: super-high efficiency (SHE), high efficiency (HE), medium efficiency (ME), low efficiency (LE), and super-low efficiency (SLE).

TE is a comprehensive evaluation index of the allocation ability and utilization efficiency of herding households' livelihood resources. TE was the lowest among the three efficiency values, with an average value of 0.762. Among them, the proportion of SLE and HE herders is relatively large, accounting for 40% and 31% of all herders, respectively. The proportion of LE and SHE pastoral households is second, accounting for 20% and 8% of all pastoral households, respectively. And the proportion of ME pastoral households is relatively small, accounting for 1% of all pastoral households. The SHE and HE pastoralists account for 39% of all pastoralists, indicating that more than one-third of pastoralists in a typical steppe of Inner Mongolia have achieved comprehensive input–output efficiency, and a small number of pastoralists have achieved ultra-high input–output efficiency. The LE and SLE herders accounted for 60% of all herders, indicating that the allocation of livelihood resources of more than half of the herders did not reach the optimal state, and the livelihood efficiency of two-thirds of the herders did not reach 0.6. Therefore, these herders still have room for improvement in the utilization and management of livelihood capital.

PTE reflects the production efficiency of input factors at the optimal scale, which is mainly affected by the organization and management ability of herders and existing technology. The PTE is the highest among the three efficiency values, with an average value of 0.955. Among them, the proportion of HE herders is the largest, accounting for 63% of all herders, and the proportions of other efficiency levels of herders are relatively small. The SHE and HE pastoralists account for 72% of all pastoralists, indicating that most pastoralists in a typical steppe of Inner Mongolia have reached the optimal state of livelihood capital utilization. However, there are still 28% herding households at a low level of livelihood efficiency, indicating that this part of herding households has not fully utilized livelihood capital input, and there is a waste of resources.

The SE reflects the gap between the actual and optimal production scale. The SE is at a lower level among the three efficiency values, with an average value of 0.801. Among them, the proportion of ME herding households is the largest, accounting for 41% of all herding households, while the proportions of LE, SLE, and HE herding households are not different, at 23%, 19%, and 17%, respectively. Only 17% of herders' livelihood efficiency reached one, indicating that less than one-fifth of herders in a typical steppe of Inner Mongolia had reached the optimal scale input and output level; more herders are at a lower-scale efficiency level. The allocation and management of livelihood capital input still needed to be adjusted and optimized.

From the perspective of return to scale (RTS), 61% of all herders had a decreasing RTS, 39% had constant RTS, and no herders had an increasing RTS, indicating that the

input–output situation of nearly half of the typical steppe herders in Inner Mongolia are basically stable, and resource allocation is in the optimal state; thus, the existing capital management strategy should be maintained. More than half of the herding households showed a diminishing RTS. For this portion of herding households, increasing livelihood capital input cannot increase livelihood output in the same proportion. Therefore, we should consider adjusting the scale of capital input and optimizing resource management to improve RTS.

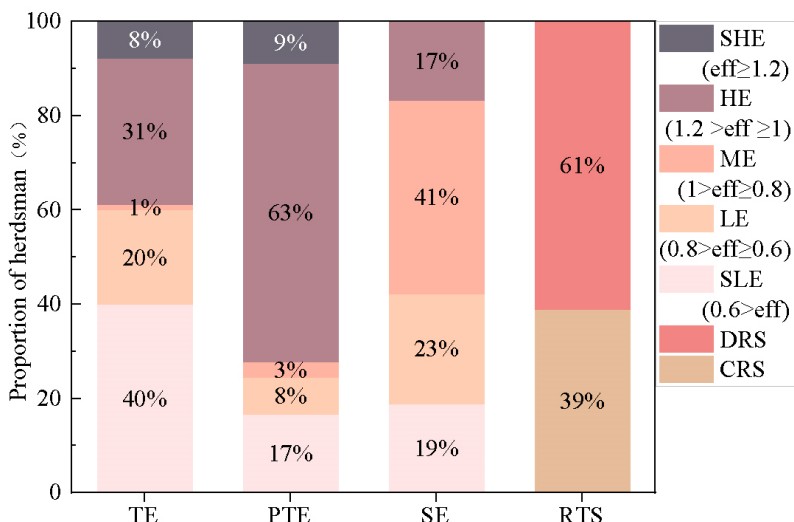

**Figure 2.** Ratio of livelihood efficiency and return to scale of herders. Note: "eff" stands for efficiency; "DRS" stands for decreasing constant returns to scale; "CRS" stands for constant return to scale.

### 3.2. Input and Output Redundancy Analysis of Herding Households

The slack variables of various input and output indices of herders' livelihoods were calculated from the SE-SBM model, and the slack variables were also called redundancy. Input redundancy represents the portion of input capital that can be reduced if the output is not being changed, and output redundancy represents the portion of the output that can be increased if the input is not being changed. When the efficiency value reaches 1, both input redundancy and output redundancy are 0; therefore, the ultra-high efficiency and high efficiency pastorals have no input or output redundancy. The redundancy of the input and output of herders whose comprehensive technical efficiency is <1 was analyzed (Figure 3), and it is found that there is room for improvement in both the inputs and outputs of herders to varying degrees.

From the perspective of input redundancy (Figure 3a), the redundancy of the human capital, the natural capital, and the information capital are low, with an average value <0.01; the redundancy of the physical capital and the financial capital are at the middle level, with an average value between 0.01 and 0.03; the redundancy of the social capital is the highest, with an average value of 0.06. This indicates that the inputs of the human capital, the natural capital, and the information capital can be reduced by herding households in a typical steppe of Inner Mongolia without reducing the output. And the inputs of the physical capital, the financial capital, and the social capital can be appropriately reduced, so as to ensure the output and simultaneously save resources.

From the perspective of output redundancy (Figure 3b), the redundancy of the income level is relatively low, with an average value of 0.38; the redundancy of the welfare level, the entertainment richness, and the happiness level is at the middle level, with an average value of 1.33–1.84; the redundancy of the living level is the highest, with an average value of 3.17. This indicates that the output of the income level can be slightly improved, and the outputs of the welfare level, the entertainment richness, and the happiness can be considerably improved. And the output of the living level can be largely improved when the inputs are

not increased. It can be observed from Figure 3 that the upward and downward buoyancy of each index of pastoral household livelihood outputs are large, and each livelihood output can be improved to varying degrees without increasing the livelihood inputs.

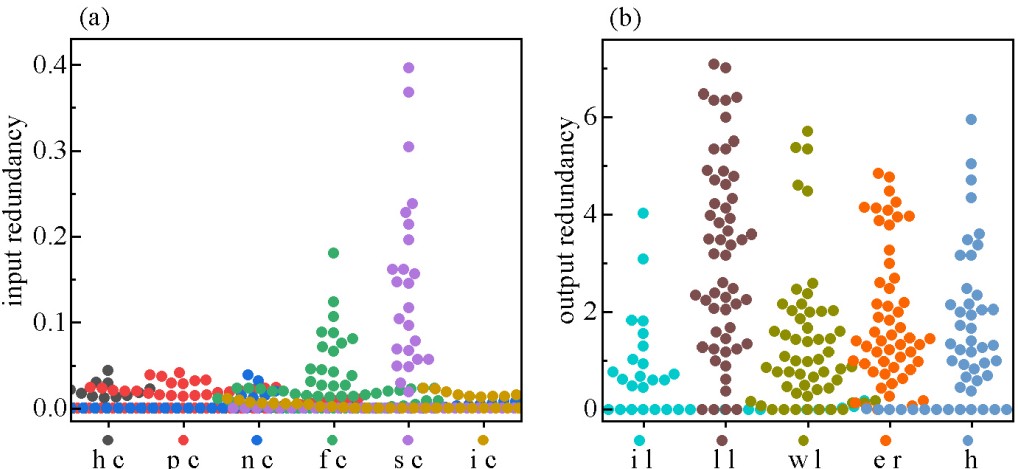

**Figure 3.** (**a**) Slack of input redundancy of herders; (**b**) slack of output redundancy of herders. Note: h c = human capital; p c = physical capital; n c = natural capital; f c = financial capital; s c = social capital; i c = information capital; i l = income level; l l = living level; w l = welfare level; e r = entertainment richness; h = happiness.

### 3.3. Factors Affecting the Livelihood Efficiency of Herders

A Tobit regression model was used to analyze the factors affecting the livelihood efficiency of typical steppe herders in Inner Mongolia. The TE of herders' livelihood was taken as the explained variable, and the factors of each herder's livelihood input, herders' animal husbandry knowledge and skills, ecological protection awareness, infrastructure, medical security, and water supply were taken as explanatory variables to analyze the influencing factors. Variables with a *p* value < 0.1 are shown Table 4.

As can be seen from Table 4, the influence coefficients of the pasture area, the loans, the access to help in times of hardship, the communication network, access to information, and the ecological protection consciousness on the livelihood efficiency of herders are significantly negative ($p < 0.05$). The degree of the infrastructure improvement, the medical security, and the water supply have significant positive effects on the livelihood efficiency of herders ($p < 0.05$). The influence coefficient of the household labor capacity and the machinery on the livelihood efficiency of herders is significantly negative ($p < 0.1$), and the influence coefficient of the livestock quantity on the livelihood efficiency of herders is significantly positive ($p < 0.1$).

This shows that there is a higher efficiency of livelihood in a typical steppe of Inner Mongolia when there is less labor capacity, less machinery, fewer loans, less access to help in times of hardship, no communication network, and less access to information for herding households. This is because these variables are all livelihood input factors. The lower the factor input, the more conducive it is to improving efficiency. The better the local infrastructure and medical security, the higher the water supply, the larger the number of livestock, and the higher the livelihood efficiency of herders. The convenience and security of life provided by society are conducive to the sustainable livelihood of herders. Natural water supply is very important in a herders' life, as it directly affects the efficiency of herders' livelihoods. The better the water supply, the higher the efficiency of the herders' livelihoods. The regression results showed that the smaller the pasture area, the larger the number of livestock, and the higher the ecological protection consciousness of herders, the lower the livelihood efficiency of herders. The large number of small livestock in the pasture area was reflected in the high stocking rate of the pasture. The higher the stocking rate of the pasture, the greater the damage to the pasture by the livestock. However, herders with

strong ecological protection consciousness pay more attention to the ecological protection of the pasture and the utilization intensity of the pasture will be lower; thus, their livelihood efficiency will be lower.

**Table 4.** Tobit regression results of influencing factors of herders' livelihood efficiency.

| Variable | Coefficient | Standard Deviation | T Value | Variable | Coefficient | Standard Deviation | T Value |
|---|---|---|---|---|---|---|---|
| Labor capacity | −0.073 | 0.038 | −1.91 * | Communication network | −0.429 | 0.089 | −4.82 *** |
| Pasture area | −0.000 | 0.000 | −4.01 *** | Access to information | −0.161 | 0.036 | −4.43 *** |
| Livestock | 0.000 | 0.000 | 1.90 * | Ecological protection consciousness | −0.053 | 0.022 | −2.40 ** |
| Machinery | −0.005 | 0.003 | −1.78 * | Infrastructure | 0.105 | 0.022 | 4.90 *** |
| Loan | −0.058 | 0.027 | −2.14 ** | Medical security | 0.055 | 0.021 | 2.61 ** |
| Access to help in times of hardship | −0.079 | 0.036 | −2.23 ** | Water supply | 0.187 | 0.040 | 4.65 *** |
| Constant term | 1.031 | 0.259 | 3.97 *** | Pseudo $R^2$ | | 2.628 | |
| Prob > chi$^2$ | | 0.000 | | Log likelihood | | 45.488 | |

Note: *** indicates that the significance level is 1%, ** indicates that the significance level is 5%, and * indicates that the significance level is 10%.

## 4. Discussion

According to Li (2017), due to the increasing marketization and improvement in production technology, the livestock production cycle in most areas of Inner Mongolia has been shortened, from the original 2-year cycle to a 1-year cycle [40]. With the shortening of the production cycle of animal husbandry, the turnover of animal husbandry is accelerated, which ultimately leads to the improvement of the production efficiency of livestock households (Wang et al., 2021) [10]. The PTE of most herders in a typical steppe of Inner Mongolia has reached comprehensive efficiency, which is consistent with previous research results. According to the law of diminishing RTS, the livelihood production scale of herders in a typical steppe of Inner Mongolia has reached a certain level, and the production efficiency of most herders' livelihood capital inputs has reached a relatively high level. Increasing the livelihood capital input cannot yield the same proportion of livelihood output. In response to this situation, herders should consider choosing the scale of operation reasonably, optimizing capital investment, seeking quality rather than quantity in livestock breeding, and shifting the large-scale breeding of livestock with small outputs to small-scale breeding of fine varieties. Namgay et al.'s (2021) research in Bhutan found that improving local cattle breeds helped herders reduce the size of their herds and adapt to a sedentary grazing lifestyle [41].

The inputs and outputs of most herders in a typical steppe of Inner Mongolia have different degrees of redundancy, where the input redundancy is small and the output redundancy is large. This indicates that the amount of livelihood capital can be reduced without reducing the livelihood output, and the herders make full use of livelihood capital. Dongdong et al. (2022) believes that the full use of livelihood capital has a positive effect on reducing the livelihood vulnerability of herders [42], and Qiu et al. (2018) believes that the increase in material capital input enhances the herders' ability to withstand natural disasters [43]. Without increasing the livelihood capital input, herders have a large space to improve various livelihood outputs and can achieve an increase in livelihood output by improving production technology, optimizing production mechanisms, and coordinating the division of labor. Birhanu's et al. (2021) research in Africa found that improving technology and management practices can increase farmers' productivity and output [44].

Affected by the increasing degradation of grassland and the grassland subsidy policy, the traditional extensive grazing mode has been changed to the semi-free and semi-enclosed farming mode in Inner Mongolia, and most of the grazing households need to buy forage to a greater or lesser extent. The large purchase of forage by herders helps to regulate the balance of forage and livestock and relieve the pressure of grassland degradation. The weight of forage purchased by herders in the study area is shown in Figure 4. Dong et al. (2023) believe that buying forage is an effective strategy for herders to adapt to extreme drought, which is conducive to improving the technical efficiency of livestock production of herders [45]. The study also showed that herders with lower livelihood efficiency had stronger awareness of ecological protection, larger pasture area, and fewer livestock, and purchased a large amount of forage to relieve the pressure of pastures; Lise et al. (2006) found that richer herding families were more concerned about environmental conditions than poor herding families [46]. Herders mainly rely on grassland resources to survive (Conte, 2015) [47], and grassland degradation will inevitably affect their sustainable livelihoods. Tiwari et al.'s (2020) study in mountainous areas of Nepal also believes that grassland degradation is one of the main factors aggravating the livelihood vulnerability of herders in mountainous areas of Nepal [48]. Only by improving the ecological protection consciousness of herders and coordinating the relationship between grassland ecology and herders' livelihoods can one obtain better development.

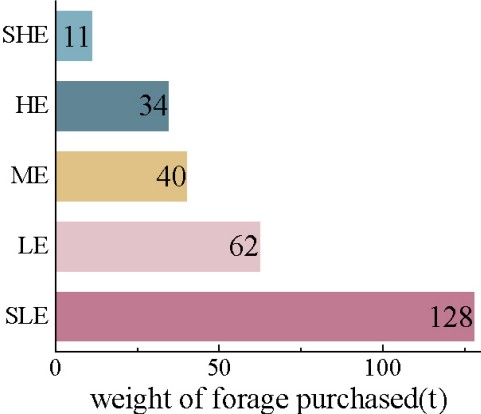

**Figure 4.** Forage purchased by herders in Inner Mongolia pastoral area.

### 5. Conclusions

Based on the interview survey data of 90 herders in a typical steppe of Inner Mongolia, this study used the SE-SBM model to construct the evaluation index system of herders' livelihood efficiency in a typical steppe of Inner Mongolia, analyzed the current situation of herders' livelihood efficiency and the improvement plan of herders' livelihood inputs and outputs, revealed the influencing factors of herders' livelihood efficiency, and draws the following conclusions:

(1) From the results of SE-SBM model, The PTE of the livelihood efficiency of herders in a typical steppe of Inner Mongolia is the highest, and the livelihood efficiency of most herders has reached one. TE and SE are relatively low, and the livelihood efficiency of most herders is low. In a typical steppe of Inner Mongolia, the RTS of most herders' livelihoods shows a decreasing state, and the livelihood input cannot obtain the same proportion of output.

(2) According to the results obtained from the SE-SBM model, the redundancy of the physical capital, the financial capital, and the social capital among the six indicators of livelihood inputs are all greater than 0.01, and the redundancy of the social capital is the highest (0.06). It shows that the outputs can be ensured and resources can be saved by appropriately reducing the inputs of the physical capital, the financial capital, and the social capital. The redundancy of the welfare level, the entertainment richness,

happiness, and the living level among the five indicators of livelihood outputs are all greater than 1.33, among which the redundancy of the living level is the highest (3.17). It shows that there is the greatest room for improvement in the living level without increasing livelihood input.

(3) According to the results of the Tobit regression model, six factors, including the pasture area, the loan, the access to help in times of hardship, the communication network, the access to information, and the ecological protection consciousness, had negative significant effects on the livelihood efficiency of herdsmen in the study area ($p < 0.05$). Particularly, the influence of the pasture area, the communication network, and the access to information is extremely significant ($p < 0.01$). Three factors, including the infrastructure, the medical security, and the water supply, had positive and significant effects on the livelihood efficiency of herders ($p < 0.05$). In particular, the infrastructure and the water supply had significant effects ($p < 0.01$).

In conclusion, this research firstly provides feasible and effective methods for evaluating the livelihood efficiency of herders in Inner Mongolia pastoral areas. Secondly, this paper comprehensively evaluates the livelihood efficiency level and influencing factors of typical steppe herdsmen in Inner Mongolia, and provides reference for future related research. Finally, it provides a reference for improving the livelihood level of herders and revitalizing herding areas in Inner Mongolia.

**Author Contributions:** Conceptualization, G.Q.; methodology, G.Q.; software, G.Q. and B.H.; formal analysis, G.Q.; resources, W.T. and S.Q.; data curation, G.Q.; writing—original draft preparation, G.Q.; writing—review and editing, G.Q., W.T. and S.Q.; visualization, G.Q.; supervision, W.T. and S.Q.; project administration, W.T. and S.Q.; funding acquisition, W.T. All authors have read and agreed to the published version of the manuscript.

**Funding:** This research was funded by the "National Nature Science Foundation of China" (funder: Wulan Tuya, grant number "41861024"), "Fundamental Research Funds for the Inner Mongolia Normal University" (funder: Wulan Tuya, grant number "2022JBZD017"), the "Natural Nature Science Foundation of Inner Mongolia" (funder: Si Qinchaoketu, grant number "2021BS04001"), and the "Science and Technology Research Project of Universities in Inner Mongolia Autonomous Region" (funder: Si Qinchaoketu, grant number "NJZZ21006").

**Institutional Review Board Statement:** Not applicable.

**Informed Consent Statement:** Informed consent was obtained from all subjects involved in the study.

**Data Availability Statement:** Not applicable.

**Acknowledgments:** The authors would like to thank the editors and anonymous reviewers for their valuable comments and suggestions which improved the quality of the manuscript.

**Conflicts of Interest:** The authors declare no conflict of interest.

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
