# Peer review of "Construction and Practice of Livelihood Efficiency Index System for Herders in Typical Steppe Area of Inner Mongolia Based on Super-Efficiency Slacks-Based Measure Model"

_sustainability, doi:10.3390/su151814005_

Round 1

Reviewer 1 Report

sustainability-2594386:Construction and practice of livelihood efficiency index system of herders in Inner Mongolia pastoral area based on the SE-SBM model

This study focuses on the livelihood efficiency of herdsmen in Inner Mongolia. Through analysis of the data collected through interviews, the authors found that the livelihood efficiency of herdsmen in Inner Mongolia is low. The efficiency of herders' livelihoods can be improved by reducing the input of livelihood factors or improving their living conditions. This is an interesting and important topic, and it is necessary to show the results to policymakers to improve management; however, there are at least several points that need to be revised in the current manuscript.

The details are as follows:

1.     Because the topic or title of the manuscript emphasizes "livelihood efficiency index system", it is necessary to organize the manuscript around the index system.

2.     In the introduction, it is necessary to introduce studies on livelihood efficiency in other areas and countries in the world, clarify the shortage in previous studies, and explain the innovation of this study.

3.     In the introduction, it is necessary to introduce other studies that adopted the SE-SBM model and similar models and explain why this study chose this model.

4.     In the materials and methods section, was the Gendar effect or variation considered during the interview? As gender may affect labor capacity significantly, it is better to add the related analysis if possible.

5.     In the Construction of the Index System, Line 178, "with reference to relevant studies", while only one reference was cited, other relevant studies should be cited.

Reviewer 2 Report

1. The format of the whole article is a little bad. Some paragraphs are not indented. The chart format is wrong. For example:

The text format in Line 183 table1 needs to be aligned.

The serial numbers of Line 249-252 Equations are not neat, and the alignment needs to be modified.

Line 459 Figure 4 needs to be modified.

2. The sentence format of the manuscript is complex. There are some grammatical problems. It is recommended to find a professional to polish the article.

3. The quality of some pictures in the manuscript is low, so it is recommended to replace them.

4. The specific contribution of the manuscript is not clear, and the reader feels more like an academic report after reading. For example, in the conclusion part of the manuscript, the authors only analyzed the current situation of herdsmen's livelihood efficiency and evaluated the influencing factors, but did not specify the academic contribution and the practical significance of the study.

5. The abstract is not closely linked to the conclusion. It is suggested that the authors modify this part.

6. In the actual collected data, it can be seen that the herdsmen surveyed are located at similar latitudes and most of them come from the same place. Can the livelihood efficiency and other indicators calculated from the survey results in this region represent the situation of the whole Inner Mongolia? The number of herders is also only 90, so the results may not be universal.

7. Please elaborate on the role of livelihood capital. Why can it be calculated as an input factor in the calculation of livelihood efficiency?

Reviewer 3 Report

I have completed the review of the manuscript Construction and practice of livelihood efficiency index system of herders in Inner Mongolia pastoral area based on the SE-SBM model, submitted to Sustainability.

Authors have used SE-SBM model to construct the evaluation index system of herders' livelihood efficiency in Inner Mongolia. The conducted work is significant in view of the livelihood proficiency of herders in the region. My major concern is that authors have used the data collected from only 90 herders. How could such a small size justify and corroborates such a huge area of the study? Further, what was the criteria for the selection of study group? These aspects are important to validate the model that has been developed.

By and large, it is Ok. Only minor corrections and editing is required

Reviewer 4 Report

The manuscript deals with an interesting piece of work, and thank you for the hard work done by the Chinese authors. However, the authors have some scope to improve the manuscript, and my comments are as follows:

  1. The abstract of the key findings says the livelihood efficiency of herdsmen in Inner Mongolia is low. What do you mean by low? It needs to explain a bit or change the word. Moreover, the general recommendation says that by reducing the input of likelihood factors or improving their livelihood conditions, it's not a recommendation; make it specify or mention what the possible inputs need to be improved.

  1. At the end of the introduction, the objectives need to be specified. Rewrite the objectives and make them simple or specific.

  1. In the evaluation index of livelihood efficiency, the human capital is missing the skill or knowledge parameters of the herding community. The skill or knowledge of herding should be included as an important parameter of human capital.

  1. The conclusion needs to be rewritten, as some portions of it are repeated in the discussion section. The conclusion should be a summary of the key findings with a future direction for the reader. It also includes a general recommendation for worldwide readers.

Good luck
